# Classifying sex and strain from mouse ultrasonic vocalizations using deep learning

**A. Ivanenko**[1,2], **P. Watkins**[3], **M. A. J. van Gerven**[4], **K. Hammerschmidt**[5], **B. Englitz**[1] *

**1** Department of Neurophysiology, Donders Institute for Brain, Cognition and Behavior, Radboud University, Nijmegen, The Netherlands, **2** Institute of Biology and Biomedicine, Lobachevsky State University, Nizhny Novgorod, Russia, **3** CAESAR, Bonn, Germany, **4** Department of Artificial Intelligence, Donders Institute for Brain, Cognition and Behavior, Radboud University, Nijmegen, The Netherlands, **5** Cognitive Ethology Laboratory, German Primate Center, Göttingen, Germany

* b.englitz@donders.ru.nl

## Abstract

Vocalizations are widely used for communication between animals. Mice use a large repertoire of ultrasonic vocalizations (USVs) in different social contexts. During social interaction recognizing the partner's sex is important, however, previous research remained inconclusive whether individual USVs contain this information. Using deep neural networks (DNNs) to classify the sex of the emitting mouse from the spectrogram we obtain unprecedented performance (77%, vs. SVM: 56%, Regression: 51%). Performance was even higher (85%) if the DNN could also use each mouse's individual properties during training, which may, however, be of limited practical value. Splitting estimation into two DNNs and using 24 extracted features per USV, spectrogram-to-features and features-to-sex (60%) failed to reach single-step performance. Extending the features by each USVs spectral line, frequency and time marginal in a semi-convolutional DNN resulted in a performance mid-way (64%). Analyzing the network structure suggests an increase in sparsity of activation and correlation with sex, specifically in the fully-connected layers. A detailed analysis of the USV structure, reveals a subset of male vocalizations characterized by a few acoustic features, while the majority of sex differences appear to rely on a complex combination of many features. The same network architecture was also able to achieve above-chance classification for cortexless mice, which were considered indistinguishable before. In summary, spectro-temporal differences between male and female USVs allow at least their partial classification, which enables sexual recognition between mice and automated attribution of USVs during analysis of social interactions.

## Author summary

Many animals communicate by producing sounds, so-called vocalizations. Mice use many different kinds of vocalizations in different social contexts. During social interaction recognizing the partner's sex is important and female mice appear to know the difference between male and female vocalizations. However, previous research had suggested that male and female vocalizations are very similar. We here show for the first time that the

**Data Availability Statement:** All raw and most processed data and code is available as collection di.dcn.DSC_620840_0003_891 and can be

downloaded at https://data.donders.ru.nl/collections/di/dcn/DSC_620840_0003_891.

**Funding:** BE was supported by a European Commission Marie-Sklodowska Curie grant (#660328; https://ec.europa.eu/research/mariecurieactions/), an NWO VIDI grant (016.189.052; https://www.nwo.nl/en/funding-our-funding-instruments/nwo/innovational-researchincentives-scheme/vidi/index.html), and a NOW grant (ALWOP.346; https://www.nwo.nl/en/news-and-events/news/2018/06/new-open-competition-acrossnwo-domain-science.html) during different parts of the project period. MG was supported by a NOW VIDI grant (639.072.513; https://www.nwo.nl/en/funding-our-funding-instruments/nwo/innovational-researchincentives-scheme/vidi/index.html). The funders had no role in study design, data collection and analysis, decision to publish, or preparation of the manuscript.

**Competing interests:** The authors have declared that no competing interests exist.

emitter's sex can be guessed from the vocalization alone, even single ones. The full spectrogram was the best basis for this, while reduced representations (e.g. basic properties of the vocalization) were less informative. We therefore conclude that while the information about the emitter's sex is present in the vocalization, both mice and our analysis must rely on complex properties to determine it. This novel insight is enabled by the use of recent machine learning techniques. In contrast, we show directly that a number of more basic techniques fail in this challenge. In summary, differences in the vocalizations between male and female mice allow to guess the emitter's sex, which enables sexual recognition between mice and automated analysis. This is important in studying social interactions between mice and how speech is produced and analyzed in the brain.

## Introduction

Sexual identification on the basis of sensory cues provides important information for successful reproduction. When listening to a conversation, humans can typically make an educated guess about the sexes of the participants. Limited research on this topic has suggested a range of acoustic predictors, mostly the fundamental frequency but also formant measures [1].

Similar to humans, mice vocalize frequently during social interactions [2–6]. The complexity of the vocalizations produced during social interactions can be substantial [7–9]. Experiments replaying male mouse courtship songs to adult females suggest that at least females are able to guess the sex of other mice based on the properties of individual vocalizations [10,11].

While in humans and other species sex-specific differences in body dimensions (vocal tract length, vocal fold characteristics) lead to predictable differences in vocalization [12,13], the vocal tract properties of male and female mice have not been shown to differ significantly [14,15]. Hence, for mice the expected differences in male/female USVs are less predictable from physiological characteristics and classification likely relies on more complex spectral properties.

Previous research on the properties of male and female ultrasonic vocalizations (USVs) in mice [16] found differences in usage of vocal types, but could not identify reliable predictors of the emitter's sex on the basis of single vocalizations.

This raises the question, whether the methods utilized were general enough to detect complex spectrotemporal differences. In several species, related classification tasks were successfully carried out using modern classifiers, e.g. hierarchical clustering of monkey vocalization types [17], or random forests for zebra finch vocalization types [18,19], non-negative matrix factorization/clustering for mouse vocalization types [20,21] or deep learning to detect mouse USVs [22], but have not addressed the task of determining the emitter's sex from individual USVs.

Here we find that the distinction of mouse (C57Bl/6) male/female USVs can be successfully performed using advanced classification using deep learning [23]: A custom-developed Deep Neural Network (DNN) reaches an average accuracy of 77%, substantially exceeding the performance of linear (ridge regression, 51%) or nonlinear (support vector machines, SVM [24], 56%) classifiers, which could be further improved to 85%, if properties of individual mice are available and can be included in the classifier.

Our DNN exploits a complex combination of differences between male/female USVs, which individually are insufficient for classification due to a high degree of within-sex variability. An analysis of the acoustic properties contributing to classification, directly shows that for most USVs only a complex set of properties is sufficient for this task. In contrast a DNN

classification on the basis of a conventional, human-rated feature set or reduced spectrotemporal properties performs much less accurately (60% and 70%, respectively). An analysis of the full network's classification strategy indicates a feature expansion in the convolutional layers, followed by a sequential sparsening and subclassification of the activity in the fully-connected layers. Applying the same classification techniques to another dataset, we can partly distinguish a (nearly) cortexless mutant strain from a wild-type strain, which had previously been considered indistinguishable on the basis of general analysis of USVs [25].

The present results indicate that the emitter's sex and/or strain can be deduced from the USV's spectrogram if sufficiently nonlinear feature combination are exploited. The ability to perform this classification provides an important building block for attributing and analyzing vocalizations during social interactions of mice. As USVs are also important biomarkers in evaluating animal models of neural diseases [26,27], in particular in social interaction, the attribution of individual USVs to their respective emitter is gaining importance.

## Results

We reanalyzed recordings of ultrasonic vocalizations (USVs) from single mice during a social interaction with an anesthetized mouse (N = 17, in 9 female awake, in 8 male awake, Fig 1A, [16]). The awake mouse vocalized actively in each recording (Male: 181±32 min⁻¹, Female: 212 ±14 min⁻¹, Fig 1B) giving a total of 10055 (female: 5723, male: 4332) automatically extracted USVs of varying spectrotemporal structure and uniquely known sex of the emitter (Fig 1C). Previous approaches for assessing the gender using basic analysis have not led to single-vocalization level predictability [16]. Here, we utilized a general framework of deep neural networks to predict the emitter's sex for single vocalizations (Fig 1D). After obtaining best-in-class performance on this challenge, we further investigate the basis for this performance. For this purpose, separate DNNs were trained for predicting features from spectrograms as well as sex from features, on the basis of human-classified features. Lastly, we analyze both the network's structure as well as the space of vocalizations in relation.

### Basic USV features differ between the sexes, but are insufficient to distinguish single USVs

While previous approaches have indicated few differences between male and female vocalizations ([16], but see [4] for CBA mice), we reassess this question first using a set of nine hand-picked features, quantifying spectrotemporal properties of the USVs (see Fig 1E and Methods for details). The first three were quantified automatically, while the latter six were scored by human classifiers.

We find significant differences in multiple features (7/9, see Fig 2A–2I for details, based on Wilcoxon Signed Ranks test) between the sexes (in all figures: red = female, blue = male). This suggests that there is exploitable information about the emitter's sex. However, the substantial variability across USVs renders each feature in isolation insufficient for classifying single USVs (see percentiles in Fig 2A–2I). Next, we investigated whether the joint set of features or the raw spectrograms have some sex-specific properties that can be exploited for classification.

Applying principal component analysis (PCA) to the 9-dimensional set of features and retaining three dimensions, a largely intermingled representation is obtained (Fig 2J). PCA of the spectrograms provides a slightly more structured spatial layout in the most variable three dimensions (Fig 2K). However, as for the features, the sexes overlap in space with no apparent separation. The basic dimensionality reduction performed by PCA, hence, reflects the co-distribution of properties and spectrograms between the sexes. At the same time it fails to even pick-up the basic properties along which the sexes differ (see above).

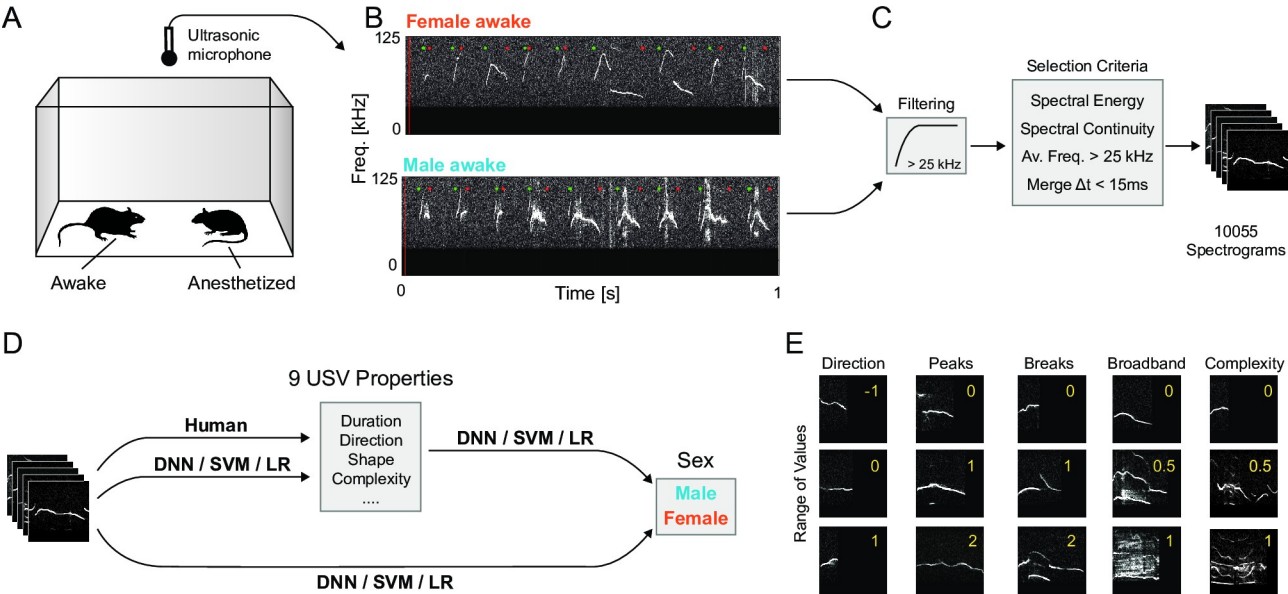

**Fig 1. Recording and classifying mouse vocalizations. A** Mouse vocalization were recorded from a pair of mice, in which one was awake, while the other was anesthetized, allowing an unambiguous attribution of the recorded vocalizations. **B** Vocalization from male and female mice (recorded in separate sessions) share a lot of properties, while differing in others. The present samples were picked at random and indicate that differences exist, while other samples would look more similar. **C** Vocalizations were automatically segmented using a set of filtering and selection criteria (see Methods for details), leading to a total set of 10055 vocalizations. **D** We aimed to estimate the properties and the sex of its emitter for individual vocalizations. First, the ground truth for the properties were established by a human classifier. We next estimated 3 relations, Spectrogram-Properties, Properties-Sex and Spectrogram-Sex directly, using both a Deep Neural Network (DNN), support vector machines (SVM) and regularized linear regression (LR). **E** The properties attributed manually to individual vocalizations could take different values (rows, red number in each subpanel), illustrated here for a subset of the properties (columns). See Methods for a detailed list and description of the properties.

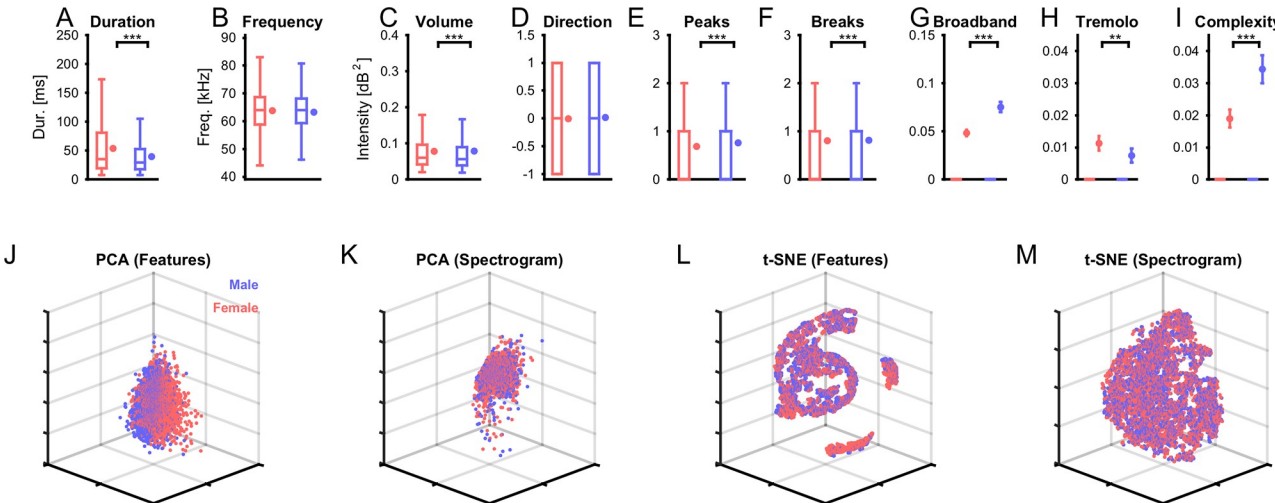

**Fig 2. Basic sex-dependent differences between vocalizations. (A-I)** We quantified a range of properties for single vocalizations (see Methods for details) and compared them across the sexes (blue: male, red: female). Most properties exhibited significant differences in median between the sexes (Wilcoxon rank sum test), except for the average frequency (**B**) and directionality (**D**). However, given the distribution of the data (box-plots, left in each panel), the variability across each property nonetheless makes it hard to use individual properties of determining the sex of the emitter. The graphs on the right for each color in each panel, show the mean and SEM. In **G-I**, only few USVs have values different than 0, hence the box-plots are sitting at 0. (**J-M**) Dimensionality reduction can reveal more complex relationships between multiple properties. We computed principal components (PCA) and t-statistic stochastic neighborhood embedding (t-SNE) for both the features (**J/L**) and the spectrograms (**K/M**). In particular, feature-based t-SNE (**L**) obtained some interesting groupings, which did, however, not separate well between the sexes (red vs. blue, see Fig 7 for more details). Each dot represents a single vocalization, after dimensionality reduction. Axes are left unlabelled, since they represent a combination of properties.

Using instead t-Distributed Stochastic Neighborhood Embedding (tSNE, [28]) for dimensionality reduction, reveals a clustered structure for both features (Fig 2L) and spectrograms (Fig 2M). For the features the clustering is very clean, and reflects the different values of the features (e.g. different number of breaks define different clusters, or gradients within clusters (Duration/Frequency)). However, USVs from different sexes are quite close in space and co-occur in the all clusters, although density-differences (of male or female USVs) exist in individual clusters. For the spectrograms the representation is less clear, but still shows much clearer clustering than after PCA. Mapping feature properties to the local clusters visible in the t-SNE did not indicate a relation between the properties and the suggested grouping based on t-SNE.

In summary, spectrotemporal features in isolation or in basic conjunction appear insufficient to reliably classify single vocalizations by their emitter's sex. However, the prior analyses do not attempt to directly learn sex-specific spectrotemporal structure from the given data-set, but instead assume a restricted set of features. Next, we used data-driven classification algorithms to directly learn differences between male and female vocalizations.

## Convolutional deep neural network identifies the emitter's sex from single USVs

Inspired by the recent successes of convolutional deep neural networks (cDNNs) in other fields, e.g. machine vision [23], we focus here on a direct classification of the emitter's sex based on the spectrograms of single USVs. The architecture chosen is a—by now—classical network structure of convolutional layers, followed by fully connected layers, and a single output representing the probability of a male (or conversely female) source (see Methods for details and Fig 3A). The cDNN performs surprisingly well, in comparison to other classification techniques (see below).

During the training procedure the cDNN's parameters are adapted to optimize performance on the test set, which was not used for training (see Methods for details on cross-validation). Performance starts out near chance level (red, Fig 3B), with the initial iterations leading to substantial changes in weights (light red). It takes ~6k iterations before the cDNN settles close to its final, asymptotic performance (here 80%, see below for average).

The learning process modifies the properties of the DNN throughout all layers. In the first layer, the neurons parameters can be visualized in the stimulus domain, i.e. similar to spatial receptive fields for real neurons. Initial random configurations adapt to become more classical local filters, e.g. shaped like local patches or lines (Fig 3C). This behavior is well documented for visual classification tasks, however, it is important to verify it for the current, limited set of auditory stimuli.

The cDNN classified single USVs into their emitter's sex at 76.7±6.6% (median±SEM, crossvalidation performed across animals, i.e. leave-one-out, n = 17 runs, Fig 3D). Classification performance did not differ significantly between male and female vocalizing mice ($p > 0.05$, Wilcoxon signed ranks test).

In comparison with more classical classification techniques, such as regularized linear regression (Ridge, blue, Fig 3E) and support vector machines (SVM, green), the DNN (red) performs significantly better (Ridge: 50.7±1.6%; SVM: 56.2±1.9%; DNN: 76.7±6.6%; all comparisons: $p < 0.01$). Shuffling the sexes of all vocalizations leads to chance performance, and thus indicates that the performance is not based on overlearning properties of a random set (Fig 3E, light colors). Instead it has to rely on the characteristic property within one sex. Further, classification is significantly better than chance for 15/17 animals (see below and S1A Fig for contributions by individual animals).

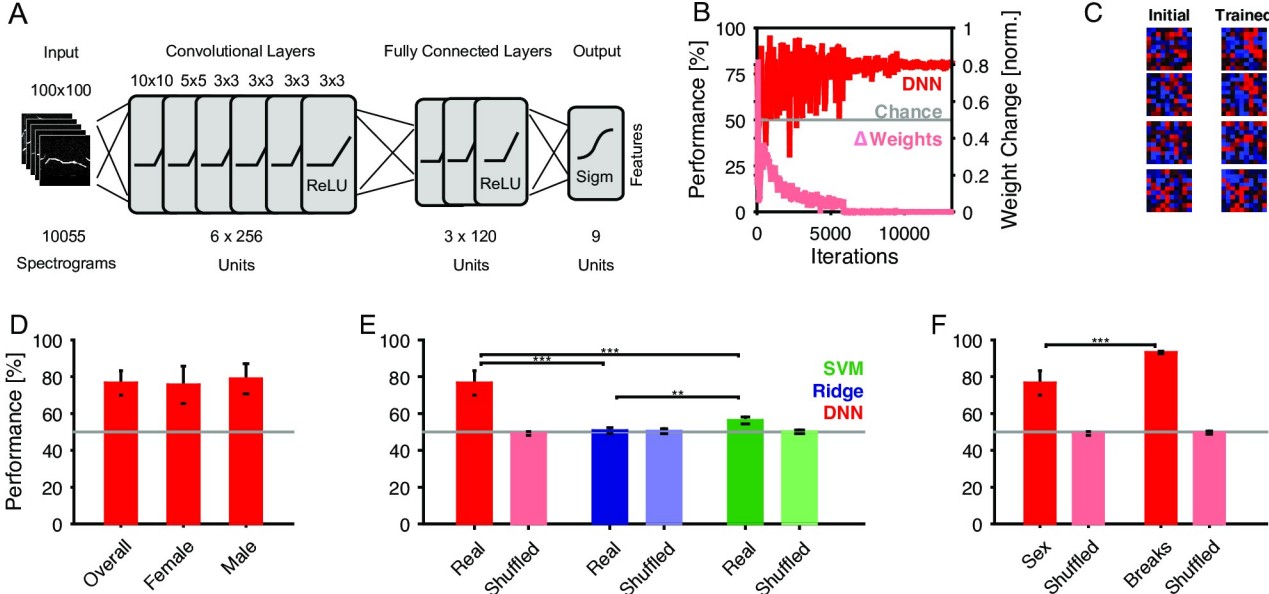

**Fig 3. Deep neural network reliably determines the emitter's sex from individual vocalizations. A** We trained a deep neural network (DNN) with 6 convolutional and 3 fully connected layers to classify the sex of emitter from the spectrogram of the vocalization. **B** The network's performance on the (Female #1) test set rapidly improved to an asymptote of ~80% (dark red), clearly exceeding chance. Correspondingly the change in the network's weights (light red) progressively decreased after stabilizing after ~6k iterations. Data shown for a representative training run. **C** The shape of the input fields in the first convolutional layer became more reminiscent of the tuning curves in the auditory system [50,51]. Samples are representatively chosen among the entire set of 256 units in this layer. **D** The average performance of the DNN (cross-validation across animals) was 76.7±6.6%, which did not differ significantly between male and female vocalizations (p>0.05, Wilcoxon rank sum test). **E** The DNN performance by far exceeded the performance of ridge regression (regularized linear regression, blue, 50.7±1.6%) and support vector machines (SVM, green, 56.2±1.9%). Bars in light colors show the corresponding estimation with randomized labels, which are all at chance level (gray line). **F** The performance by the DNN was not only limited by the properties of the spectrograms (e.g. background noise, sampling, etc.) since a DNN trained on the number of breaks (right bars) performed significantly better. This control shows that the identical set of stimuli can be better separated on a simpler (but also binary) task. Light bars again show performance on randomized labels.

Lastly, to verify that the general properties of the spectrograms do not pose a sex-independent limit on classification performance, we evaluated the performance of the same network on another complex task: to determine whether a vocalization has a break or not. On the latter task the network can do significantly better than on classifying the emitter's sex, reaching 93.3% (p<0.001, Wilcoxon signed ranks test, Fig 3F).

Altogether, the estimated cDNN clearly outperforms classical competitors on the task and reaches a substantial performance, which could potentially be further improved, e.g. by the addition of more data for refining the DNN's parameters.

## Including individual vocalization properties improves classification accuracy

The network trained above did not have access to properties of vocalization of individual mice. We next asked to what degree individual properties can aid the training of a DNN to improve the attribution of vocalizations to an emitter. This was realized by using a randomly selected set of USVs for crossvalidation, such that the training set contained USVs from every individual.

The resulting classification performance increases further to 85.1±2.9% (median across mice), and now all mice are recognized better than chance (S1B Fig). We also tested to which degree the individual properties suffice for predicting the identity of the mouse instead of sex (see Methods for details on the output). The average performance is 46.4±7.5% (median across

animals, S1C Fig), i.e. far above chance performance (5.9% if sex is unknown, or at most 12.5% if the sex is known), which is also reflected in all of the individual animals (M3 has the largest p-value at 0.0002, binomial test against chance level).

Hence, while not fully identifiable, individual mice appear to shape their vocalizations in characteristic ways, which contributes to the increased performance of the DNN trained using crossvalidation on random test sets (S1B Fig). It should be emphasized, however, that information on the individual vocalization properties will not be available in every experiment and thus this improvement has only limited applicability.

## USV features are insufficient to explain cDNN performance on sex classification

Simple combinations of features were insufficient to identify the emitter's sex from individual USVs, as demonstrated above (Fig 2). However, it could be that more complex combinations of the same features would be sufficient to reach similar performance as for the spectrogram-based classification (Fig 3). We address this question by predicting the sex from the features alone—as classified by a human—using another DNN (see Methods for details and below). Further, we check whether the human-classified features can be learned by a cDNN. Together, these two steps provide a stepwise classification of sex from spectrograms.

Starting with the second point, we estimated separate cDNNs for 4 of the 6 features, i.e. direction, number of breaks, number of peaks and spectral breadth of activation ('broadband'). The remaining two features—tremolo and complexity—were omitted, since most vocalizations scored very low in these values, thus creating a very skewed training set of the networks. A near optimal—but uninteresting—learning outcome is then a flat, near zero classification. The network structure of these cDNNs was chosen identical to the sex-classification network, for simpler comparison (Fig 4A).

Average performance for all four features was far above chance (Fig 4B–4E, maroon, respective chance levels shown in gray), with some variability across the different values of each property (Fig 4B–4E, red). This variability is likely a consequence of the distribution of values in the overall set of USVs (Fig 4B–4E, light blue, scaled also in percentage) in combination with their inherent difficulty in prediction as well as higher variability in human assessment. Except for the broadband classification (82.0±0.4%), the classification performance for the features (Direction: 73.0±0.5%; Breaks: 68.6±0.4%; Peaks: 57.0±1.0%) stayed below the one for sex (76.7±0.9%).

Next, we estimated a DNN without convolutional layers for predicting the emitter's sex from a basic set of 9 features (see Methods for details, and Fig 4F). The overall performance of the DNN was above chance (59.6±3.0%, Fig 4G, maroon), but remained well below the full spectrogram performance (76.7%). The DNN performed similarly on these features as SVM (57.5±3.7%) and ridge regression (61.7±3.0%) (Fig 4H), suggesting that the non-convolutional DNN on the features did not have a substantial advantage, pointing to the relevance of the convolutional layers in the context of the spectrogram data. For ridge regression, the contribution of the different features in predicting the emitter's sex can be assessed directly (Fig 4I), highlighting the USVs' duration, volume and spectral breadth as the most distinguishing properties (compare also [4]). Volume appears surprising, given that the relative position between the microphone and the freely moving mouse was uncontrolled, although there may still be an average effect based on the sex-related differences in vocalization intensity or head orientation.

Finally, we investigated whether describing each USV by a more encompassing set of extracted features would allow higher prediction quality. In addition to the 9 basic features above we included 15 additional extracted features (see Methods for full description) and in

particular certain spectrotemporal, 1D features of vocalizations. Briefly, for the latter we chose the fundamental frequency line (frequency with maximal intensity per time point in spectrogram, dim = 100), the marginal frequency content (dim = 233), the marginal intensity

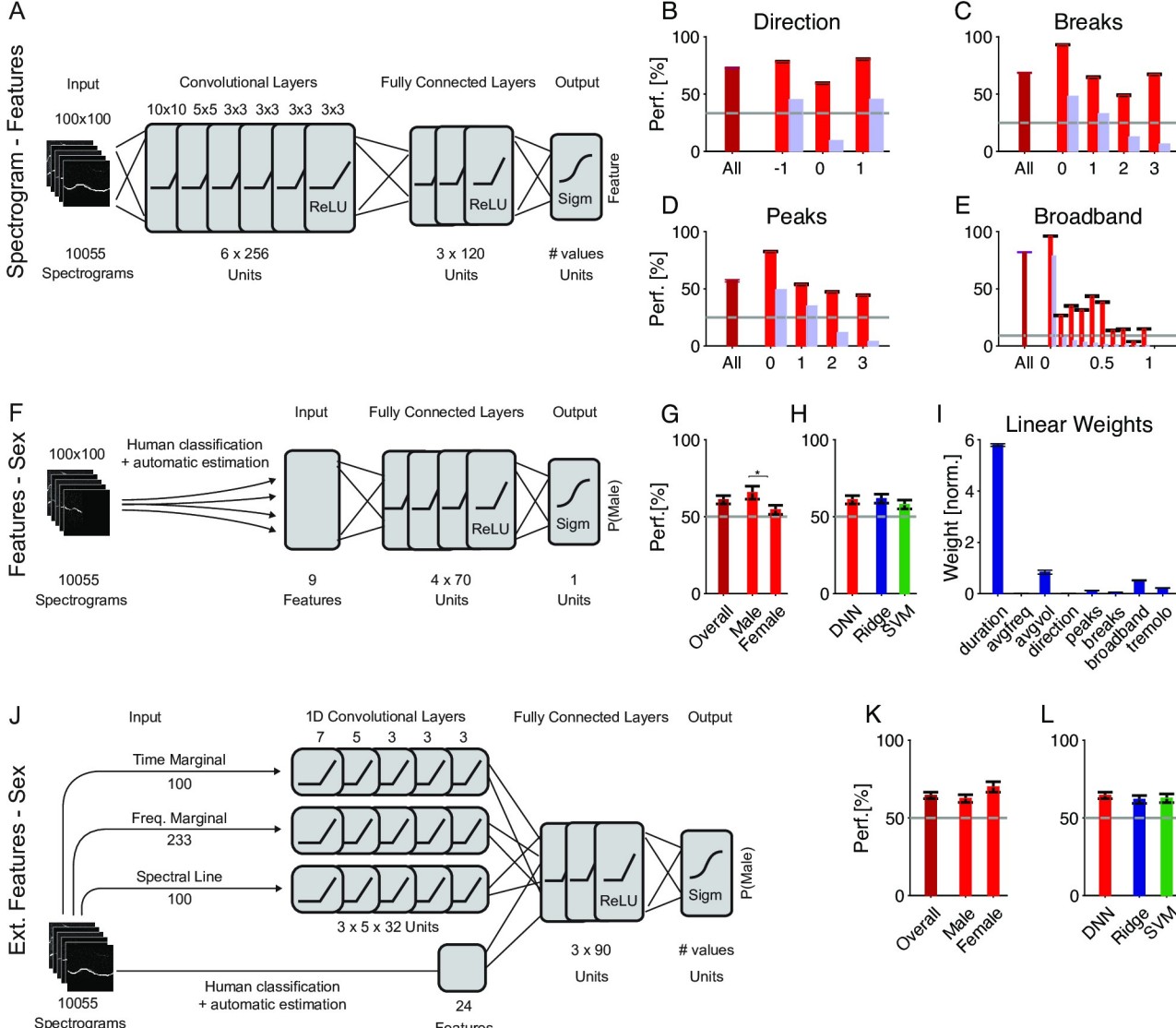

**Fig 4. Features alone are insufficient to explain the DNN classification performance. A** Features of individual vocalizations can also be measured using dedicated convolutional DNNs, one per feature, with identical architecture as for sex classification (see Fig 3A). **B-E** Classification performance for different properties was robust, ranging between 57.0 and 82.0% on average (maroon) and depending on the individual value of each property (red). We trained networks for direction ({-1,0,1}, **B**), the number of breaks ({0–3}, **C**), the number of peaks ({0–3}, **D**) and the degree of broadband activation ([0,1], **E**). For the other 2 properties (complex and tremolo), most values were close to 0 and thus networks did not have sufficient training data for these. The light gray lines indicate chance performance, which depends on the number of choices for each property. The light blue bars indicate the distributions of values, also in %. **F** Using a non-convolutional DNN, we investigated how predictable features alone would be, i.e. without any information about the precise spectral structure of each vocalization. **G** Prediction performance was above chance (maroon, 59.6±3.0%) but less than the prediction of sex on the basis of the raw spectrograms (see Fig 3). The gray line indicates chance performance. **H** Feature-based prediction of sex with DNNs performed similarly compared to ridge regresson (blue) and SVM (red, see main text for statistics). **I** Duration, volume and the level of broadband activation were the most significant linear predictors for sex, when using ridge regression. **J** Using a semi-convolutional DNN, we investigated the combined predictability of the same features as above, plus 3 statistics of the stimulus (each a vector), i.e. the marginal of the spectrogram in time and frequency, as well as the spectral line, i.e. the sequence of frequencies of maximal amplitude per time-bin. **K** The average performance of the semi-convolutional DNN (64.5%) stays substantially lower than the 2D cDNN (see Fig 3D). USVs of both sexes were predicted with similar accuracy. **L** The average performance of the semi-convolutional DNN is not significantly larger than ridge regression (61.9%) or SVM (62.7%) on the same data, due to the large variability across the sexes (see Panel **K**).

progression (dim = 100). Together with a total of 24 extracted, single value properties (see Methods for list and description), each USV was described by a 457 dimensional vector. For the DNN, the three 1D properties were fed each into a separate 1D convolutional stack (see Methods for details), and then combined with the 24 features as input to the subsequent fully connected layers. We refer to this network as a semi-convolutional DNN.

The performance of this network (64.5±2.1%) was significantly higher than the basic feature network (59.3±3.0%, p<0.05, Wilcoxon signed ranks test), however, did not reach the performance of the full spectrogram network. We also ran corresponding ridge regression and SVM estimates for completeness, whose performance remained on average below the semi-convolutional DNN, which was, however, nonsignificant.

While the estimation of certain features is thus possible from the raw spectrograms, their predictiveness for the emitter's sex stays comparatively low for both simple and advanced prediction algorithms. While the inclusion of further spectrotemporal features improved the performance, it still stayed below the cDNN performance for full spectrograms. A sequential combination of the two networks—i.e. raw-to-feature followed directly by feature-to-gender—would perform worse than either of the two. Hence, we hypothesize that the direct sex classification from spectrograms must rely on a different set of local or global features, not well captured in the present set of features.

## Classification of different strains

For social interactions, typically same strain animals are used. However, in understanding the neural basis of USV production, mutant animals are of great value. Recently, [25] analyzed social vocalizations of *Emx1-CRE;Esco2* mice, which lack the hippocampus and nearly all of cortex. Contrary to expectation, they concluded that their USVs did not different significantly, questioning the role of cortex in the patterning of individual USVs. We reanalyzed their dataset using the same DNN-architecture as described above, finding significant differences between WT and mutant mice on the basis of their USVs (63.4±5.3% correct excluding individual properties (Fig 5A), again outpacing linear and nonlinear classification methods (Ridge regression (blue), 51.0±1.5%, p = 0.017) and support vector machines (SVM, green, 55.0±4.1%, p = 0.089, i.e. close, but not significant at the p<0.05 level), Fig 5B). Including individual properties again

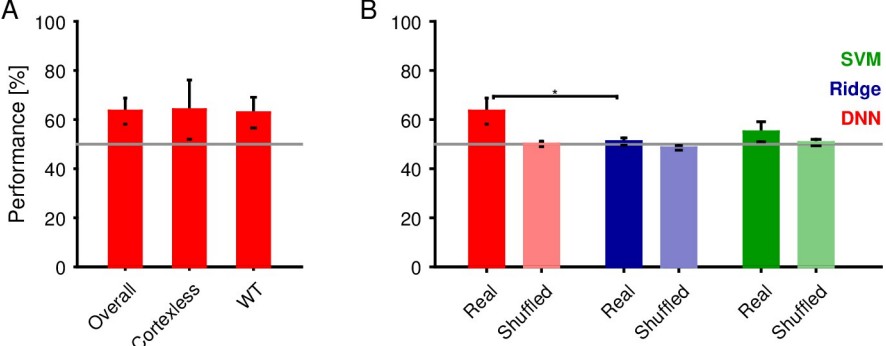

**Fig 5. Deep neural network partly determines the emitter's strain from individual vocalizations.** We trained a deep neural network (DNN) with 6 convolutional and 3 fully connected layers to classify the strain (WT vs. cortexless) from the spectrograms of each vocalization (see Fig 3A for a schematic of the network structure). **A** The average performance of the DNN (cross-validation across recordings) was 63.4±5.3%. WT vocalizations were classified with an accuracy that did not differ statistically. **B** The DNN performance exceeded the performance of ridge regression (regularized linear regression, blue, 51.0±1.5%) and support vector machines (SVM, green, 55.0.2±4.1%). Bars in light colors show the corresponding estimation with randomized labels, which are all at chance level (gray line).

improved performance substantially (74.2% with individual properties, i.e. via random test-sets, S2A Fig), although this may often not be a desirable or feasible option.

While the classification is not as clear as between male/female vocalizations, we suggest that the previous conclusion regarding the complete lack of distinguishability between WT and *Emx1-CRE;Esco2* needs to be revisited.

## Separation of sexes and representation of stimuli increases across layers

While the sex classification cDNN's substantial improvement in performance over alternative methods is remarkable (Fig 3), we do not have direct insight into the internal principles that it applies to achieve this performance. We investigated the network's structure by analyzing its activity and stimulus representation across layers ('deconvolution', [29]), using the tf_cnnvis package [30]. Briefly, we performed two sets of analyses: (i) the across-layer evolution of neuronal activity for individual vocalizations in relation to gender classification, and (ii) the stimulus representation across and within layers in relation to gender classification, for details see Methods.

First, we investigated the patterns of neuronal activation as a function of layer in the network (Fig 6A–6D). We illustrate the representation for two randomly drawn vocalizations (Fig 6A top: Female example; bottom: Male example). In the convolutional layers (Fig 6B), the activation pattern transitions from a localized, stimulus-aligned representation to a coarsened representation, following the progressive neighborhood integration in combination with the stride (set to [2,2,1,1,1] across the layers). As the activation reaches the fully connected layers, all spatial relations to the stimulus are discarded.

The sparsity of representation across layers of the network increased significantly, going from convolutional to fully connected (Fig 6C, Female: red; Male: blue; sparsity was computed as 1 - #[active units]/#[total units], ANOVA, p indistinguishable from 0, for n's see # of USVs in Methods). This finding bears some resemblance with cortical networks, where higher level representations become sparser (e.g. [31]). Also, the correlation between activation patterns of same-sex vocalizations increased strongly and significantly across layers (Fig 6D, red and blue). Conversely, the correlation across different-sex activation patterns became significantly more negative, in particular in the last fully connected layer (Fig 6D, purple). Together, these correlations form a good basis for classification as male/female, by the weights of the output layer (top, right).

Second, we investigated the stimulus representation as a function of layer (Fig 6E–6H). The representation of the original stimulus (Fig 6E) became successively more accurate (see Fig 6G below) stepping through the layers of the network (Fig 6F, left to right). While lower layers exhibited still some ambiguity regarding the detailed shape, the stimulus representation in higher layers reached a plateau around convolutional layer 4/5, at 0.4 for males and ~0.46 for females (Fig 6G). This correlation appears lower than indicated visually, however, some contamination remained around the vocalization. Across neurons in a layer, this representation stabilized across layers, reaching a near-identical representation on the highest level (Fig 6H).

In summary, both the separation of the sexes and the representation of the stimuli improved as a function of the layer, suggesting a step-wise extraction of classification relevant properties. While the convolutional layers appear to mostly expand the representation of the stimuli, the fully connected layers then become progressively more selective for the final classification.

## Complex combination of acoustic properties identifies emitter's sex

Lastly, it would be insightful to understand differences between male and female vocalizations on the level of their acoustic features. As the original space of the vocalizations is high-

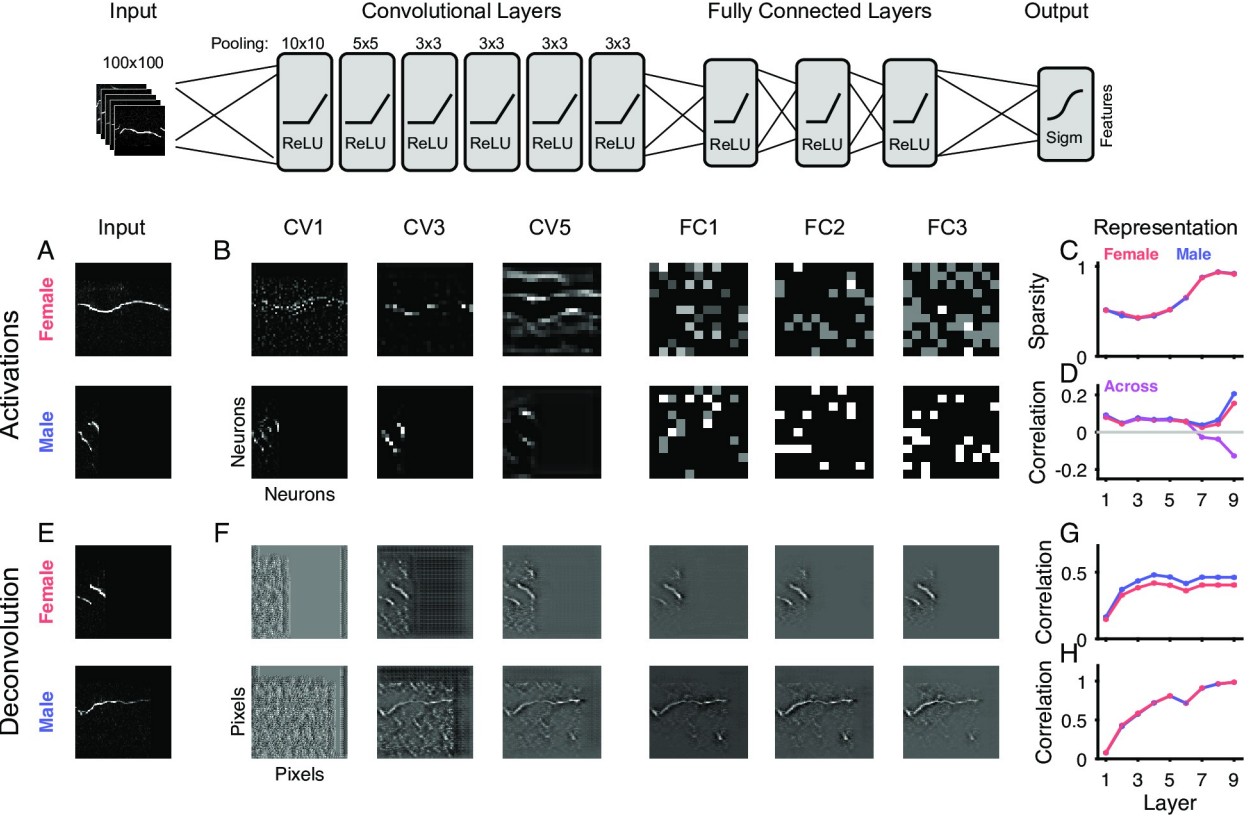

**Fig 6. Structural analysis of network activity and representation.** Across the layers of the network (top, columns in B/F) the activity progressively dissociated from the image level (compare A and B) (left, A/E). The stimuli (**A**, samples, top: female; bottom: male) are initially encoded spatially in the early convolutional layers (**B**, left, CV), but progressively lead to more general activation. In the fully connected layers (**B**, right, FC), the representation becomes non-spatial. Concurrently, the sparsity (**C**) and within-sex correlation between FC representations increases (**D**, red/blue) towards the output layer, while across-sex correlation decreases (**D**, purple). The average correlation, however, stays limited to ~0.2, and thus the final performance is only achieved in the last step of pooling onto the output units. Using deconvolution [29], the network representation was also studied across layers. The representation of individual stimuli (**E**) became more faithful to the original across layers (**F**, from left to right, top: female, bottom: male sample). Correlating the deconvolved stimulus with the original stimulus exhibited a fast rise to an asymptotic value (**G**). In parallel, the similarity of the representation between the neurons of a layer improved through the network (**H**). In all plots in the right column, the error bars indicate 2 SEMs, which are vanishingly small due to the large number of vocalizations/neurons each point is based on.

dimensional (~10000 given our resolution of the spectrogram), we performed a dimensionality reduction using t-SNE [28] to investigate the structure of the space of vocalizations and their relation to emitter sex and acoustic features.

The vocalizations' t-SNE representation in three dimensions exhibits both particular large scale structure as well as some local clustering (Fig 7A). Male (blue) and female (red) vocalizations are not easily separable but can form local clusters or at least exhibit differences in local density. The most salient cluster is formed by male vocalizations (Fig 7A, bottom left), which contains almost no female vocalizations. Examples from this cluster (Fig 7B, bottom three) are similar in appearance and differ from vocalizations outside this cluster (Fig 7B, top two). Importantly, vocalizations in this cluster do not arise from a single male, but all male mice contribute to the cluster.

The local density of male and female USVs already appears different on the large scale (Fig 7C, top), mostly due to the lack of the male cluster (described above). Taking the difference in local density between male and female USVs (Fig 7C, bottom), shows that differences also exist throughout the overall space of vocalizations (indicated by the presence of red and blue

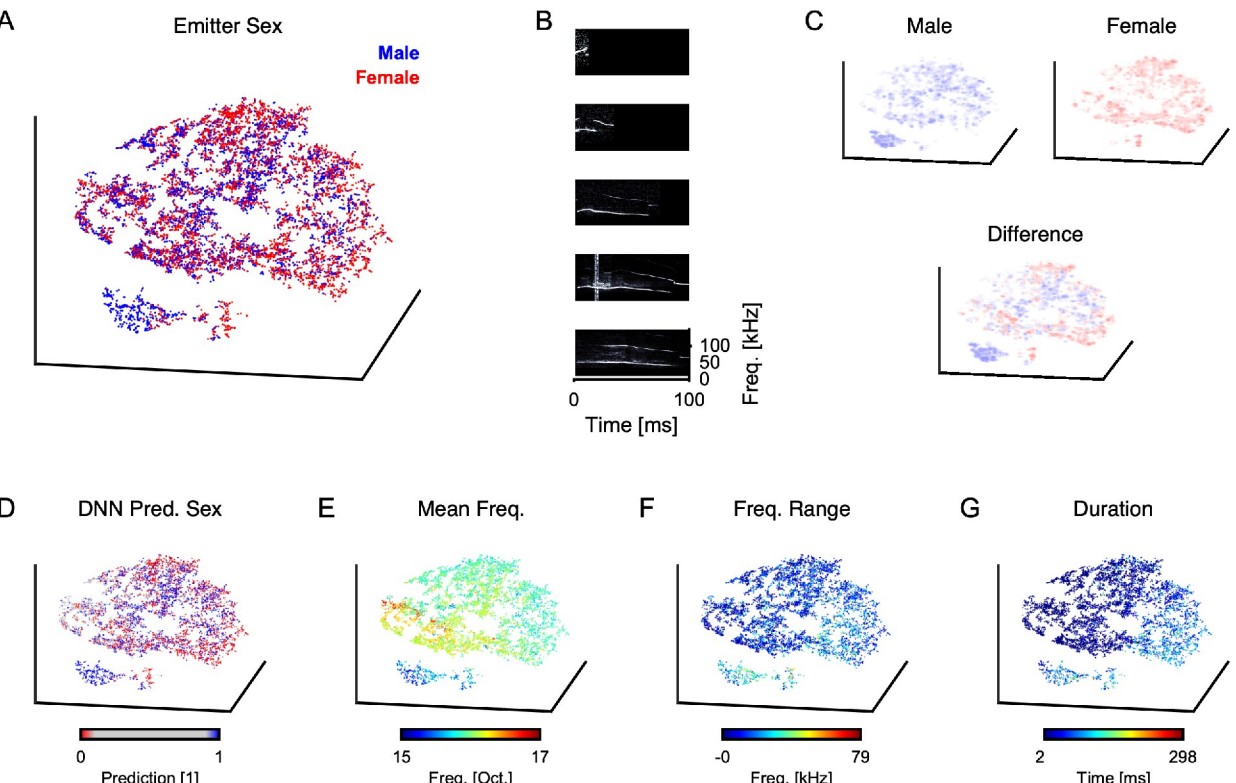

**Fig 7. In-depth analysis of vocalization space indicates complex combination of properties distinguishing emitter sex. A** Low-dimensional representation of the entire set of vocalizations (t-SNE transform of the spectrograms from 10^4 to 3 dimensions) shows limited clustering and structuring, and some separation between male (blue) and female (red) emitters. See also S1 Video, which is a dynamic version of the present figure, revolving all plots for clarity. **B** Individual samples of vocalizations, where the bottom three originate from the separate, large male cluster in the lower left of **A**. They all have a similar, long low-frequency call, combined with a higher-frequency, delayed call. The male cluster contains vocalizations from all male mice, and is hence not just an individual property. This indicates that a subset of vocalizations is rather characteristic for its emitter's sex. **C** The difference (bottom) between male (top left) and female (top right) densities indicates interwoven subregions of space dominated by one sex, i.e. blue subregions indicate male-dominant vocalization types, and red subregions female dominant. **D** Restricting to a subset of clearly identifiable vocalizations (based on DNN output certainty, <0.1 (female) and >0.9 (male)) provides only limited improvement in separation, indicating that the DNN decides based on a complex combination of subregions/spectrogram properties. **E** Mean frequency of the vocalization exhibits local neighborhoods on the tSNE representation, in particular linking the dominantly male cluster with exceptionally low frequencies. **F** Similarly, the frequency range of vocalizations in the dominantly male cluster is comparably high. **G** Lastly, the typical duration of the dominantly male cluster lies in a middle range, while not being exclusive in this respect.

regions, which would not be present in the case of matched densities). These differences in local density can be the basis for classification, e.g. performed by a DNN.

Performing nearest neighbor decoding [32] using leave-one-out crossvalidation on the t-SNE representation yielded a prediction accuracy of 62.0%. This indicates that the dimensionality reduction performed by t-SNE is useful for the decoding of emitter sex, while not nearly sufficient to attain the performance of the DNN on the same task.

The classification performed by the DNN can help to identify typical representatives of male and female vocalizations. For this purpose we restrict the USVs to those that the DNN assigned as 'clearly female' (<0.1, red) and 'clearly male' (>0.9, blue), which leads to slightly better visual separation and significantly better nearest neighbor decoding (Fig 7D, 66.8%).

Basic acoustic properties could help to explain both the t-SNE representation as well as provide an intuition to different usage of USVs between the sexes. Among a larger set tested, we here present mean frequency (Fig 7E), frequency range (Fig 7F) and duration (Fig 7G). Mean frequency shows a clear neighborhood mapping on the t-SNE representation. The male cluster

appears to be dominated by low average frequency, and also a wide frequency range and a mid-range duration (~100 ms).

Together, we interpret this to indicate that the DNNs classification is largely based on a complex combination of acoustic properties, while only a subset of vocalizations is characterized by a limited set of properties.

## Discussion

The analysis of social communication in mice is a challenging task, which has gained traction over the past years. In the present study we have built a set of cDNNs which are capable of classifying the emitter's sex from single vocalizations with substantial accuracy. We find that this performance is dominated by sex-specific features, but individual differences in vocalization are also detectable and contribute to the overall performance. The cDNN classification vastly outperforms more traditional classifiers or DNNs trained on predefined features. Combining tSNE based dimensionality reduction with the DNN classification we conclude that a subset of USVs is rather clearly identifiable, while for the majority the sex rests on a complex set of properties. Our results are consistent with the behavioral evidence that mice are able to detect the sex on the basis of single vocalizations.

### Comparison with previous work on sex differences in vocalization

Previous work has approached the task of differentiating the sexes from USVs using different techniques and different datasets. The results of the present study are particularly interesting, as a previous, yet very different analysis of the same general data set did not indicate a difference between the sexes [16]. The latter automatically calculated 8 acoustic features for each USV and applied clustering analysis with subsequent linear discriminant analysis. Their analysis identified 3 clusters and the features contributing most to the discrimination were duration, frequency jumps and change. We find a similar mapping of features to clusters, however, both Hammerschmidt et al. [16] and ourselves did not find salient differences relating to the emitter's sex using these analysis techniques, i.e. either for dimensionality reduction (Fig 2) and linear classification (Fig 3).

Extending the approach of using explicitly extracted parameters, we tested a substantially larger set of properties (see Methods), describing each vocalization in 457 dimensions. The properties were composed of 6 human-scored features, XXX automatically extracted composite features, and the fundamental frequency line and its intensity marginals. The above set was classified using both regression, SVM and a semi-convolutional DNN. We would have expected this large set of properties to sufficiently describe each USV, and thus allow classification at similar accuracy with the semi-convolutional DNN as the full DNN. To our surprise this was not the case, leading to a significantly lower performance (Fig 4). We hypothesize that the full DNN picks up subtle differences or composite properties that were not captured by our hand-designed set of properties, despite its breadth, and inclusion of rather general properties as the fundamental frequency line and its intensity marginals.

While the human scored properties only formed a small subset of the full set of properties, we note that human scoring has its own biases (e.g. variable decision threshold through the lengthy classification process, ~60h > 1 week).

Previous studies on other data sets have led to mixed results, with some studies finding differences in USV properties [33], others finding no structural differences but only on the level of vocalization rates [16,34]. However, to our knowledge, no study was so far able to detect differences on the level of single vocalizations, as in the present analysis.

## Comparison to previous work on automated classification of vocalizations

While there does not exist much advanced work on sexual classification using mouse vocalizations, there has been considerable advances in other vocalization-based classification tasks. Motivating our choice of technique for the present task, an overview is provided below for the tasks of USV detection, repertoire analysis, and individual identification (the latter not for mice).

Several commercial (e.g. Avisoft SASLab Pro, or Noldus UltraVox XT) and non-commercial systems (e.g. [7]) were able to perform USV detection with various (and often not systematically tested) degrees of success. Recently, multiple studies have advanced the state of the art, in particular DeepSqueak [22], based also on DNNs. The authors performed a quantitative comparison of the performance and robustness of USV detection, suggesting that their DNN approach is well suited for this task and superior to other approaches (see Fig 2 in [22]).

Both DeepSqueak and another recent package MUPET [20] include the possibility to extract a repertoire of basic vocalization types. While DeepSqueak converges to a more limited, less redundant set of basic vocalization types, the classification remains non-discrete in both cases. This apparent lack of clear base categories, may be reflected in our finding of a lack of simple distinguishing properties between male and female USVs. Nonetheless, MUPET was able to successfully differentiate different mouse strains using their extracted repertoires. Other techniques for classification of USV types include SVMs and Random Forest classifiers (e.g. [35]), although the study does not provide a quantitative comparison between SVM and RF performance.

In other species, various groups have successfully built repertoires or identified individuals using various data analysis techniques. Fuller [17] used hierarchical clustering to obtain a full decomposition of the vocal repertoire of Male Blue Monkeys. Elie & Theunissen [18] applied LDA directly to PCA-preprocessed spectrogram data of zebra finch vocalization to assess their discriminability. Later the same group [19] applied LDA, QDA and RF to assess the discriminability of vocalizations among individuals. While these results are insightful in their own right, we think the degree of intermixing in the present data-set would not allow linear or quadratic techniques to be successful, although it would be worthwhile to test RF based classification on the present dataset.

## Contributions of sex-specific and individual properties

While the main focus of the present study was on identifying the sex of the emitter from single USVs, the results also demonstrate that individual differences in vocalization properties contribute to the identification of the sex. Previous research has been conflicting on this issue, with a recent synthesis proposed that suggests a limited capability for learning and individual differentiation [36]. From the perspective of classifying the sex, this is undesired, and we therefore checked to which degree the network takes individual properties into account when classifying sex. The resulting performance when testing the network only on entirely unused individuals is lower (~77%), but it is well known that it will underestimate the true performance (since a powerful general estimator will tend to overfit the training set, e.g. [37]). Hence, if we expand our study to a larger set of mice, we predict that the sex-specific performance would increase to around ~80 (assuming that the performance is roughly between the performances for 'individual properties used' and 'individual properties interfering'). More generally, however, the contribution of individual properties provides an interesting insight in its own right, regarding the developmental component of USV specialization, as the animals are otherwise genetically identical.

## Optimality of classification performance

The spectrotemporal structure of male and female USVs could be inherently overlapping to some degree and hence preclude perfect classification. How close the present classification comes to the best possible classification is not easy to assess. In human speech recognition, scores of speech intelligibility or other classifications can be readily obtained (using psycho-physical experiments) to estimate the limits of performance, at least with respect to humans, often considered the gold standard. For comparison, human sex can be obtained with very high accuracy from vocalizations (DNN-based, 96.7%, [38]), although we suspect classification was based on longer audio segments than those presently used. For mice, a behavioral experiment would have to be conducted, to estimate their limits of classifying the sex or identity of another mouse based on its USVs. Such an estimate would naturally be limited by the inherent variability of mouse behavior in the confines of an experimental setup. In addition, the present data did not include spatial tracking, hence, our classification may be hampered by the possibility that the animal switched between different types of interactions with the anesthetized conspecific.

## Biological versus laboratory relevance

Optimizing the chances of mating is essential for successfully passing on one's genes. While there is agreement that one important function of mouse vocalizations is courtship [2,3,6,34], to which degree USVs contribute to the identification of a conspecific has not been fully resolved, partly due to the difficulty in attributing vocalizations to single animals during social interaction. Aside from the ethological value of deducing the sex from a vocalization, the present system also provides value for laboratory settings. The mouse strain used in the present setting is an often used laboratory strain, and in the context of social analysis the present set of analysis tools will be useful, in particular in combination with new, more refined tools for spatially attributing vocalizations to individual mice [4].

## Generalization to other strains, social interactions and USV sequences

The present experimental dataset was restricted to only two strains and two types of interaction. It has been shown previously that different strains [20,39] and different social contexts [2,21,33] influence the vocalization behavior of mice. As the present study depended partially on the availability of human-classified USV features, we chose to work with more limited sets here. While not tested here, we expect that the performance of the current classifier would be reduced if directly applied to a larger set of strains and interactions. However, retraining a generalized version of the network, predicting both sex, strain and type of interaction would be straightforward and is planned as a follow-up study. In particular the addition of different strains would allow the network to be used as a general tool in research.

A potential generalization would be the classification of whole sequences ('bouts') of vocalizations, rather than the present single vocalization approach. This could help to further improve classification accuracy, by making more information and also the inter-USV periods available to the classifier. However, in particular during social interaction, there may be a mixture of vocalizations from multiple animals in close temporal succession, which need to be classified individually. As this is not known for each sequence of USVs, a multilayered approach would be necessary, i.e. first classify a sequence, and if the certainty of classification is low, then reclassify subsequences until the certainty per sequence is optimized.

### Towards a complete analysis of social interactions in mice

The present set of classifiers provides an important building block for the quantitative analysis of social vocalizations in mice and other animals. However, there remain a number of generalizations to reach its full potential. Aside from adding additional strains and behavioral contexts, the extension to longer sequences of vocalizations is highly relevant. While we presently demonstrate that single vocalizations already provide a surprising amount of information about the sex and individual, we agree with previous studies that sequences of USVs play an important role in mouse communication [2,7,40]. While neither of these studies differentiated between the sexes, we consider a combination of the two approaches an essential step of further automatizing and objectifying the analysis of mouse vocalizations.

## Methods

We recorded vocalizations from mice and subsequently analyzed their structure to automatically classify the mouse's sex and spectrotemporal properties from individual vocalizations. All experiments were performed with permission of the local authorities (Bezirksregierung Braunschweig) in accordance with the German Animal Protection Law. Data and analysis tools relevant to this publication are available to reviewers (and the general public after publication) via the Donders repository (see Data Availability Statement)

### Data acquisition

The present analysis was applied to two datasets, one recording from male/female mice and one from cortex-deficient mutants, described below. The latter is covered as supplementary material to keep the presentation in the main manuscript easy to follow. Both datasets were recorded using AVISOFT RECORDER 4.1 (Avisoft Bioacoustics, Berlin Germany) sampled at 300 kHz. The microphone (UltraSoundGate CM16) was connected to a preamplifier (UltraSoundGate 116), which was connected to a computer.

### Male/Female experiment

The recordings constitute a subset of the recordings collected in a previous study [16]. Briefly, C57BL/6NCrl female and male mice (>8w), were housed in groups of five in standard (Type II long) plastic cages, with food and water ad libitum. The resident-intruder paradigm was used to elicit ultrasonic vocalizations (USVs) from male and female 'residents'. Resident mice (males and females) were first habituated to the room: Mice in their own home cage were placed on the desk in the recording room for 60 seconds. Subsequently, an unfamiliar intruder mouse was placed into the home cage of the resident, and the vocalization behavior was recorded for 3 min. Anesthetized females were used as 'intruders' to ensure that only the resident mouse was awake and could emit calls. Intruder mice were anaesthetized with an intraperitoneal (i.p.) injection of 0.25% tribromoethanol (Sigma-Aldrich, Munich, Germany) in the dose 0.125 mg/g of body weight. Overall, 10055 vocalizations were recorded from 17 mice. Mice were only recorded once, such that 9 female and 8 male mice contributed to the data set. Male mice produced 542±97 (Mean ± SEM) vocalizations, while female mice produced 636 ±43 calls over the recording period of 3 min.

### WT/Cortexless paradigm

The recordings are a subset of those collected in a previous study [25]. Briefly, each male mouse was housed in isolation one day before the experiment in a Macrolon 2 cage. During the recordings, these cages were placed in a sound-attenuated Styrofoam box. After 3 minutes,

a female (Emx1-CRE;Esco2fl/fl) was introduced in the cage with the male and the vocalizations recorded for 4 minutes.

Overall, 4427 vocalizations were recorded from 12 mice (6 WT and 6 *Emx1-CRE;Esco2*).

## Data processing and extraction of vocalizations

An automated procedure was used to detect and extract vocalizations from the continuous sound recordings. The procedure was based on existing code [7], but extended in several ways to optimize extraction of both simple and complex vocalizations. The quality of extraction was manually checked on a randomly chosen subset of the vocalizations. We here describe all essential steps of the extraction algorithm for completeness (the code is provided in the repository alongside this manuscript, see above).

The continuous recording was first high-pass filtered (4th order Butterworth filter, 25 kHz) before transforming it into a spectrogram (using the norm of the fast fourier transform for 500 samples (1.67 ms) windows using 50% consecutive window overlap, leading to an effective spectrogram sampling rate of ~1.2 kHz). Spectrograms were converted to a sparse representation by setting all values below the 70th percentile to zero (corresponding to a 55±5.2 dB threshold below the respective maximum in each recording), which eliminated most close-to-silent background noise bins. Vocalizations were identified using a combination of multiple criteria (defined below), i.e. exceeding a certain spectral power, maintaining spectral continuity, and lie above a frequency threshold. Vocalizations that followed each other at intervals <15 ms were subsequently merged again and treated as a single vocalization. For later classification, each vocalization was represented as a 100×100 matrix, encoded as uint8. Hence, vocalizations longer than 100 ms (11.7%) were truncated to 100 ms.

The three criteria above were defined as follows:

- *Spectral energy*: The distribution of spectral energies was computed across the entire spectrogram, and only time-points were kept in which any of the bins exceeded 99.8% of the distribution (manually estimated). While this threshold seems high, we verified manually that this threshold did not exclude any clearly recognizable vocalizations. It reflects the relative rarity of bins containing energy from a vocalization.

- *Spectral continuity*: We tracked the maximum position across frequencies for each time-step, and computed the size of the difference in frequency location between time-steps. These differences were then accumulated, centered at each time-point for 3.3 ms (4 steps), in both directions. Their minimum was compared to a threshold of 15, which corresponds to ~3.4 kHz/ms. Hence, the vocalization was accepted if the central line did not change too much. The threshold was set generously, in order to also include more broadband vocalizations. Manual inspection indicated that no vocalizations were missed due to a too stringent spectral purity criterion.

- *Frequency threshold*: Despite the high-pass filtering some low-frequency environmental noises can contaminate higher frequency regions. We then excluded all vocalizations whose mean spectral energy was <25 kHz.

Only if these three properties were fulfilled, we included a segment of the data into the set of vocalizations. The segmentation code is included in the above repository (MATLAB Code: VocCollector.m), with the parameters set as follows: SpecDiscThresh = 0.8, MergeClose = 0.015.

## Dimensionality reduction

For initial exploratory analysis we performed dimensionality reduction on the human-classified features and the full spectrograms. Both principal component analysis (PCA) and t-

statistic neighborhood embedding (tSNE, [28]) were applied to both datasets (see Fig 2). tSNE was run in Matlab using the original implementation from [28], using the following parameters (initial reduction using PCA to dimension 9 for features and 100 for spectrograms; perplexity: 30).

## Classification

We performed automatic classification of the emitter's sex, as well as several properties of individual vocalizations (see below). While the sex was directly known, the ground truth for most of the properties was estimated by two independent human evaluators, who had no access to the emitting sex during scoring.

**Vocalization properties.** For each vocalization we extracted a range of properties on multiple levels, which should serve to characterize each vocalization at different scales of granularity. The code for assigning these properties is provided in the above repository (MATLAB Code: VocAnalyzer.m). Prior to extraction of the properties, the vocalization's spectrogram was filtered using a level set curvature flow technique [41] using an implementation available online by Pragya Sharma (github.com/psharma15/Min-Max-Image-Noise-Removal). This removes background noise while leaving the vocalization parts nearly unchanged. We also only worked with the spectrogram >25 kHz to deemphasize background noise from movement related sound, while keeping all USVs, which—to our experience—are never below 25 kHz for male-female social interactions. Below the spectrogram of a USV is referred to as $S(f,t) = |STFT(f,t)|^2$, where $STFT$ denotes the short-term fourier transform.

- *Fundamental Frequency Line*: Mouse USVs essentially never show harmonics, since the first harmonic would typically be located very close or above the recorded frequency range (here: 125kHz). Hence, at a given time, there is only one dominant frequency in the spectrogram. We next used an edge detection algorithm (MATLAB: edge(Spectrogram,'Prewitt')) to locate all locations of the spectrogram where there is a transition from background to any sound. Next, the temporal range of the USV was limited to time-bins where at least one edge was found. The frequency of the fundamental for each point in time was then identified as the frequency bin with maximal amplitude. If no edge was identified between two bins with an edge, the corresponding bins' frequencies were set to 0, indicating a break in the vocalization. If the frequency in a single bin between two adjacent bins with an edge differed by more than 5kHz, the value was replaced by the interpolation (this can occur if the amplitude inside a fundamental is briefly lowered). The fundamental frequency line is a 1-dim. function referred to as *FF(t)*. For input to the DNN, was *FF(t)* shortened or lengthened to 100 ms, the latter by adding zeros.

- *Fundamental Energy Line*: The value of $S(FF(t),t)$ for all $t$ for which $FF(t)$ is defined and >0. The fundamental amplitude line is a 1-dim. function referred to as *FE(t)*. Since the USVs only have a single spectral component per time, *FE(t)* is here used as the temporal marginal.

- *Spectral Marginal*: The spectral marginal was computed as the temporal average per frequency bin of *FE(t)*, where *FE(t)>0*. To make it a useful input for a convolutional DNN, the same, complete frequency range was used for each USV, which had a dimension of 233.

- *Spectral Width*: defined as *SW = max(FF(t))—min(FF(t))*. This estimate was chosen over conventional approaches using the spectral marginal, to focus on the USV and ignore surrounding noise.

- *Duration*: time *T* from first to last time-bin of the spectral line (including time-bins with frequency equal to 0 in between).

- *Starting Frequency*: *FF(0)*.

- *Ending Frequency*: *FF(T)*, where *T* is the length of *FF*.

- *Minimal Frequency*: min(*FF(t)*), computed over all *t* where *FF(t)>0*).

- *Maximal Frequency*: max(*FF(t)*), computed over all *t* where *FF(t)>0*).

- *Average Frequency*: we included two estimates here, (1) the average frequency of *FF(t)*, computed over all *t* where *FF(t)>0*) and (2) the intensity-weighted average across the raw spectrogram, computed as $\frac{1}{T}\sum_{t=1}^{T}\sum_{f=1}^{F}F(f)*S(f,t)/\sum_{f=1}^{F}S(f,t)$

- *Temporal Skewness*: skewness of *FE(t)*.

- *Temporal Kurtosis*: kurtosis of *FE(t)*.

- Spectral Skewness: skewness of the spectral marginal (i.e. across frequency)

- *Spectral Kurtosis*: kurtosis of the spectral marginal (i.e. across frequency)

- *Direction*: the sign of the mean of single-step differences of *FF(t)* for neighboring time-bins, for all *t* where *FF(t)>0*.

- *Spectral Flatness (Wiener Entropy)*: The Wiener Entropy WE [42] was computed as the temporal average of the Wiener Entropy for each time-bin of the USV, i.e.: $WE(S) = \frac{1}{N}\sum_{t=1}^{T}\frac{G(S(f,t))}{<S(f,t)>_f}$, where G denotes the geometric mean, i.e. $G(S) = (\prod_{f=Fmin}^{Fmax}S(f,t))^{1/n}$.

- *Spectral Salience*: Spectral salience was computed as the temporal average of the ratios between the largest, off-zero peak and the peak at 0 of the autocorrelation across frequencies for each time bin.

- *Tremolo*: i.e. whether there is a sinusoidal variation in frequency of *FF(t)*. To assess the tremolo we applied the spectral salience estimate to *FF(t)*.

- *Spectral Energy*: the average of *FE(t)*.

- *Spectral Purity*: average of the instantaneous spectral purity, defined as $SP(S)) = \frac{1}{N}\sum_{t=1}^{T}(max_f(S(f,t)/median_f(S(f,t)))$

  In addition, the following 6 properties were also scored by two human evaluators:

- *Direction*: whether the vocalization is composed mostly of flat, descending or ascending pieces, values are 0,-1,1, respectively

- *Peaks*: number of clearly identifiable peaks, values: integers $\geq 0$.

- *Breaks*: number of times the main fundamental frequency line is disconnected, values: integers $\geq 0$

- *Broadband*: whether the frequency content is narrow (0) or broadband (1). Values: [0,1]

- *Tremolo*: whether there is a sinusoidal variation on the main whistle frequency. Values: [0,1]

- *Complexity*: whether the overall vocalization is simple or complex. Values: [0,1]

**Performance evaluation.** The quality of classification was assessed using cross-validation, i.e. by training on a subset of the data (90%) and evaluating the performance on a separate test set (remaining 10%). For significance assessment, we divided the overall set into 10 non-

overlapping test sets (permutation draw, with their corresponding training sets), and performed 10 individual estimates. Performance was assessed as percent correct, i.e. the number of correct classifications in comparison with the total number of the same type. We performed another control, where the test set was just one recording session, which allowed us to verify that the classification was based on sex rather than individual (see Fig 4B). Here, the size of the test set was determined by the number of vocalizations of each individual.

## Deep neural network classification

For classification of sexes, features and individuals, we implemented several neural networks of standard architectures containing multiple convolutional and fully connected layers (DNNs, see details below). The networks were implemented and trained in Python 3.6 using the TensorFlow toolkit [43]. Estimations were run on single GPUs, a NVIDIA GTX1070 (8 GB) and a NVIDIA RTX2080 (8 GB).

The networks were trained using standard techniques, i.e. regularisation of parameters using batch-normalization [44] and dropout [45]. Batch normalization was applied for all network layers, both convolutional and fully connected. The sizes and the number of features for convolutional kernels were selected according to the parameters commonly used for natural images processing networks (specific architectural details are displayed in each figure). For training, stochastic optimization was performed using ADAM [46]. The rectified linear unit (ReLU) was used as the activation function for all layers except for the output layer. For the output layer, the sigmoid activation function was used. The cross entropy loss function was used for minimization. The initial weight values of all layers were set using Xavier initialization [47].

## Spectrogram-to-Sex, Spectrogram-to-Cortexless, Spectrogram-to-Individual and Spectrogram-to-Features Networks

In total, 8 convolutional networks taking spectrograms as their input were trained: *Spectrogram-to-Sex*, Spectrogram-to-Cortexless, 2 *Spectrogram-to-Individual (for 'cortexless' and 'gender' individual data sets)* and 4 *Spectrogram-to-Features* networks including direction (*Spectrogram-to-Direction*), number of peaks (*Spectrogram-to-Peaks*), number of breaks (*Spectrogram-to-Breaks)* and broadband property (*Spectrogram-to-Broadband*) detection networks.

The networks consisted of 6 convolutional and 3 fully connected layers. The detailed layer properties are represented in Table 1. The output layer of *Spectrogram-to-Sex* network

**Table 1. The architecture of Spectrogram-to-Sex and Spectrogram-to-Features networks.**

| Layer | Properties |
| --- | --- |
| Convolutional 1 | Kernel size: 10x10, 256 units, stride = 2 |
| Convolutional 2 | Kernel size: 5x5, 256 units, stride = 2 |
| Convolutional 3 | Kernel size: 3x3, 256 units, stride = 1 |
| Convolutional 4 | Kernel size: 3x3, 256 units, stride = 1 |
| Convolutional 5 | Kernel size: 3x3, 256 units, stride = 1 |
| Convolutional 6 | Kernel size: 3x3, 256 units, stride = 1 |
| Fully connected 1 | 120 (80*) units |
| Fully connected 2 | 120 (80*) units |
| Fully connected 3 | 120 (80*) units |

\* For Spectrogram-to-Broadband feature network we used 80-unit fully connected layers

**Table 2. The architecture of Feature-to-Sex network.**

| Layer | Size |
|---|---|
| Fully connected 1 | 70 units |
| Fully connected 2 | 70 units |
| Fully connected 3 | 70 units |
| Fully connected 4 | 70 units |

contained single element, representing the detected sex (the probability of being male). The *Spectrogram-to-Direction*, *Spectrogram-to-Peaks* and *Spectrogram-to-Breaks* networks' output layers contained the number of elements equal to the number of detected classes. Since only a few samples had a number of peaks or breaks more than 3, the data set was restricted to 4 classes: no peaks (breaks), 1 peak (break), 2 peaks (breaks), 3 or more peaks (breaks). So, the output layers of *Spectrogram-to-Peaks* and *Spectrogram-to-Breaks* networks consisted of 4 elements representing each class. The *Spectrogram-to-Broadband* network output layer contained a single element representing *Broadband* value.

## Features-to-Sex network

The networks used for classification of sexes based on the spectrogram features consisted of 4 fully connected layers. The first layer received inputs from 9-dimensional vectors representing feature values: duration, average frequency, average volume, direction. number of peaks, number of breaks, "broadband" value, "vibrato" value, "complex" value (the latter 6 were determined by human classification). The output layer contained single element, representing the detected sex (the probability of being male). The detailed layer properties are represented in Table 2.

## Extended features-to-Sex network

This emitter-sex classification network combines convolutional and non-convolutional processing steps: features were directly input to the fully-connected layers, while separate convolutional layer-stacks were trained for three 1D quantities, i.e. the marginal spectrum (233 dimensional), the marginal intensity over time (100 dimensional) and the maximum frequency line (100 data points). The convolutional sub-networks otherwise had an identical architecture with 5 layers (see Fig 4J and Table 3). Their outputs were combined with all 24 discrete acoustic feature data (see above) to form the input to the subsequent, fully connected, 3 layer sub-network. The output layer contained a single element, representing the detected sex (the probability of being male).

**Table 3. The architecture of Extended Features-to-Sex network.** Note that the network contains 3 parallel convolutional sub-networks of the architecture described in the table.

| Layer | Properties |
|---|---|
| Convolutional 1 | Kernel size: 7, 32 units, stride = 2 |
| Convolutional 2 | Kernel size: 5, 32 units, stride = 2 |
| Convolutional 3 | Kernel size: 3, 32 units, stride = 1 |
| Convolutional 4 | Kernel size: 3, 32 units, stride = 1 |
| Convolutional 5 | Kernel size: 3, 32 units, stride = 1 |
| Fully connected 1 | 90 units |
| Fully connected 2 | 90 units |
| Fully connected 3 | 90 units |

**Table 4. *Spectrogram-to-Sex* and *Spectrogram-to-Individual* networks training protocol.**

| Stage | Epochs | Learning rate | Dropout probability |
|-------|--------|---------------|---------------------|
| 1 | 40 | 0.001 | 0.0 |
| 2 | 25 | 0.0001 | 0.5 |
| 3 | 25 | 0.00001 | 0.7 |

## Input data preparation and augmentation

The original source spectrograms were represented as N×233 images where N is the rounded duration of the spectrogram in milliseconds. The values were in the range [0,255]. To represent the spectrograms as 100×100 images they were cut to 100 ms threshold and rescaled to M×100 size keeping the original aspect ratio, M is the scaled thresholded duration. The rescaled matrix was aligned to the left side of the resulting image.

For DNNs, we implemented on-the-fly data augmentation to enlarge the input dataset. For each 2D spectrogram image being fed to the network during training session a set of modifications were applied including start and end times clip (up to 10% percent of original duration), intensity (volume) amplification with a random coefficient drawn from the range [0.5, 1.5] and the addition of gaussian noise with the variance randomly drawn from the range [0, 0.01]. The same algorithm of augmentation was applied for 1D spectral line data and time marginal data (*Extended-Features-To-Gender network*), with no amplification. For marginal frequency data the augmentation included intensity amplification with coefficient drawn from the range [0.8, 1.2] and gaussian noise only. We used the scikit-image package [48] routines for implementing data augmentation operations.

Different approaches were used to compensate for the asymmetry in the occurrence of different classes (e.g. for there were 32% more female than male vocalizations), we used different procedures for the three classification methods: For Ridge and SVM, the number of male and female vocalizations was equalized in the training sets by reducing the size of the bigger sets. For DNNs, a loss-function was used with weighting according to the occurrence in the different classes.

## Training protocols

For *Spectrogram-to-Sex*, *Spectrogram-to-Individual* and the training protocol included 3 stages with different training parameters set. (Table 4). For *Spectrogram-to-Features* and *Features-to-Gender* the training protocols included 2 stages (Tables 5 and 6). *Extended-Features-to-Gender* network was trained in 4 stages(Table 7). Batch size was set to 64 samples for all networks.

## Analysis of activation patterns and stimulus representation

The network's representation as a function of layer was analyzed on the basis of a standard deconvolution library *tf_cnnvis* [30], implementing activity and representation ('deconvolution') analysis introduced earlier [29]. Briefly, the deconvolution analysis works by applying the transposed layer weight matrix to the output of the convolutional layer (feature map),

**Table 5. *Spectrogram-to-Features* networks training protocol.**

| Stage | Epochs | Learning rate | Dropout probability |
|-------|--------|---------------|---------------------|
| 1 | 40 | 0.001 | 0.0 |
| 2 | 25 | 0.0001 | 0.5 |

**Table 6.** *Features-to-Gender* network training protocol.

| Stage | Epochs | Learning rate | Dropout probability |
|---|---|---|---|
| 1 | 150 | 0.001 | 0.0 |
| 2 | 200 | Quadratic decay from 0.001 to $10^{-5}$ | 0.5 |

taking into account strides and padding settings. Applied sequentially to the given and all preceding layers, it produces an approximate reconstruction of the original image for each neuron, allowing to relate particular parts of the image to the set of features detected by the layer.

The routines provided by the library were integrated into the sex classification code to produce activation and deconvolution maps for all spectrograms of the dataset, which were then subjected to further analysis (see Fig 6).

For the activations, we computed the per-layer sparsity (i.e. fraction of activation per stimulus, separated by emitter sex) as well as the per-layer correlation between activation patterns within and across emitter sex. The latter serves as an indicator of the degree of separation between the sexes (Fig 6, middle).

For the deconvolution results (i.e. approximations of per-neuron stimulus representation), we computed two different correlations: First, the correlation of each neurons representation with the actual stimulus, again separated by layer and sex of the emitter. Second, the correlation of between the representation of neurons within a layer, across all layers and sex by the emitter (Fig 6, bottom).

## Linear regression and support vector machines

To assess the contribution of basic linear prediction to the classification performance, we performed regularized (*ridge*) regression by direct implementation of the normal equations in MATLAB. The performance was generally close to chance and did not depend much on the regularization parameter.

To check whether simple nonlinearities in the input space could account for the classification performance of the DNN we used support vector machine classification [24,49], using their implementation in MATLAB (*svmtrain*, *svmclassify*, *fitcsvm*, *predict*). We used the quadratic kernel for the spectrogram-based classification, and the linear (dot-product) kernel for the feature-based classification. The performance was above chance, however, much poorer than for DNN classification.

For both methods the evaluation was done just as for the DNN estimation, by using single individual test sets excluded from the training data (see Figs 3 and 5 for comparative results).

## Statistical analysis

Generally, nonparametric tests were used to avoid distributional assumptions, i.e. Wilcoxon's rank sum test for two group comparisons, and Kruskal-Wallis for single factor analysis of variance. When data were normally distributed, we checked that statistical conclusions were the

**Table 7.** *Extended-Features-to-Gender* network training protocol (single individual test set).

| Stage | Epochs | Learning rate | Dropout probability |
|---|---|---|---|
| 1 | 50 | 0.001 | 0.0 |
| 2 | 100 | 0.0001 | 0.3 |
| 3 | 100 | 0.00001 | 0.5 |
| 4 | 100 | 0.000002 | 0.5 |

same for the corresponding test, i.e. t-test or ANOVA, respectively. Effect sizes were computed as the variance accounted by a given factor, divided by the total variance. Error bars indicate 1 SEM (standard error of the mean). Post-hoc pairwise multiple comparisons were assessed using Bonferroni correction. All statistical analyses were performed using the statistics toolbox in MATLAB (The Mathworks, Natick).

## Supporting information

**S1 Fig. Classification based on individual properties can support classification of sex. A** Comparing the performance of the network across individual indicates that the sex-classification is better than chance in 15/17 animals. The light gray line indicates chance, and the dark gray line average performance. **B** If individual properties are included in the classification (through the use of random testset during crossvalidation, 10x), overall performance increases to 85.1% (median across animals). **C** We trained another DNN to classify mouse identity, achieving above chance performance for almost all mice, indicating directly that differences between individual mice can also contribute to the classification of sex. For classification of individuals, the chance level is at $100/N_{mice}$%.
(TIF)

**S2 Fig. Same layout as S1 Fig.** 1 for the classification of wild-type vs. cortexless animals.
(TIF)

**S1 Video. Three-dimensional representation of the entire set of vocalizations (t-SNE transform of the spectrograms from 10^4 to 3 dimensions) shows limited clustering and structuring, and some separation between male (blue) and female (red) emitters.**
(MP4)

## Acknowledgments

The authors would like to thank Hugo Huijs for manually classifying the entire set of vocalizations.

## Author Contributions

**Conceptualization:** B. Englitz.

**Formal analysis:** A. Ivanenko, P. Watkins, B. Englitz.

**Investigation:** A. Ivanenko.

**Resources:** B. Englitz.

**Software:** A. Ivanenko, B. Englitz.

**Supervision:** B. Englitz.

**Validation:** A. Ivanenko.

**Visualization:** B. Englitz.

**Writing – original draft:** A. Ivanenko, P. Watkins, M. A. J. van Gerven, K. Hammerschmidt, B. Englitz.

**Writing – review & editing:** A. Ivanenko, M. A. J. van Gerven, K. Hammerschmidt, B. Englitz.

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
