## [Decision Letter · Decision Letter 0]

4 Dec 2019

Dear Dr Ivanenko,

Thank you very much for submitting your manuscript 'Classification of mouse ultrasonic vocalizations using deep learning' for review by PLOS Computational Biology. Your manuscript has been fully evaluated by the PLOS Computational Biology editorial team and in this case also by independent peer reviewers. The reviewers appreciated the attention to an important problem, but raised some substantial concerns about the manuscript as it currently stands. While your manuscript cannot be accepted in its present form, we are willing to consider a revised version in which the issues raised by the reviewers have been adequately addressed. We cannot, of course, promise publication at that time.

Sincerely,

Frédéric E. Theunissen

Associate Editor

PLOS Computational Biology

Lyle Graham

Deputy Editor

PLOS Computational Biology

[LINK]

Dear Alexander and Bernhard,

Both I and the reviewers found that your use of the DNN for classifying SUV is interesting - although I still find it somewhat incremental. Both reviewers have addressed minor points that you should address. I am very much in agreement with reviewer 1 that the only cross-validation test that is correct is what you call the single recording test set. Your paper should not use any other tests - period! This is true not only for the DNN but with your comparisons with other methods like linear regression. Other wise you are admitting that you have a "random- effect" the identity of each mouse but ignoring it. It is as if you knew that you need to do mixed-effect modeling but refrain from doing so. In terms of the analysis of vocalizations, this point was very explicitly made by Mundry and Sommer (Anim Behavior 2007) as applied to linear classifiers - they call their approach permuted DFA. Note also that this is the type of cross-validation that my lab has always performed both with Hyenas and with Zebra Finches. I have also not looked at what is usually done for mouse for USV but the 9 bio acoustical features you choose are quite simple - I would like you try a much more complete set of features that include a better description of the spectral enveloppe (spectral mean, quartiles, etc), temporal envelope and fundamental frequencies (we have used such features in our work but they are quite commonly used by bioacoustibcians). For the fundamental frequency, you might even to PC of the time-varying fundamental (eg. Dahl A, Sherlock BR, Campos JJ, Theunissen FE, 2014.

Best wishes,

Frederic.

Reviewer's Responses to Questions

**Comments to the Authors:**

Reviewer #1: Review: PCOMPBIOL-D-19-01759

Ivanenko et. Al., have developed several DNNs (Deep Neural Networks) capable of classifying individual USVs into discrete categories with accuracy not previously achievable. Most notably they are capable of classifying the sex of the emitting mouse ~78% of the time. They were also able to extend this technique to classify USVs from a transgenic mouse, as well as classifying USVs by sever experimenter defined acoustic properties. The work reported here is extremely thorough and described in excellent detail, and the underlying method could prove extremely useful for researchers hoping to analyze vocalizations during opposite sex interactions. Computational methods of identification have significant benefits over competing methods, such as multi-microphone arrays, namely zero cost and ease of implementation. Although Dr. Englitz appears to be working on that solution in parallel, as well as considering combining the methods in the future. The technique also has some drawbacks. This is only useful for studying opposite-sex interactions, and 78% accuracy may be too low to effectively study syllable ordering, or syntax. The precise pattern of vocalizations is thought to be an important feature of social vocalization, and miss-categorizing 22% of syllables could drastically change results.

I have no major concerns

Minor concerns and suggestions

Calling the network 84% accurate is a bit misleading. DNNs are really meant to classify new information, and the network is only 78% accurate when tested against a novel mouse. Although the authors admit this, they lean heavily on the 84% number throughout the manuscripts.

The title feels a little broad, and perhaps should include classification of sex or genotype.

This manuscript seems to have two main threads. The first is proof that information about the sex of emitter mice is contained within individual USVs. I believe the authors do an excellent job proving this point, and they show clearly that the sex information is embedded in a complex combination of many features, including individual variation in USV production. While this work is thorough and well done, my enthusiasm for it is not particularly high. The features underlying classification are still unknown and I’m not exactly sure why it would be useful for mice identify sex based on a single call. It feels like this thread occupies the bulk of the text and sort of occludes the authors more important contribution.

Namely, the authors have laid the groundwork for a universal sex classifier for mouse USVs. The current network is ~78% accurate within novel animals of the same strain, a major improvement to all existing methods I am aware of. The great thing about neural networks is they aren’t fixed or permanent. They can be shared and retrained by researchers around the world who have already collected datasets from separate strains or transgenic animals. Unfortunately, I couldn’t find the final networks among the available data. The Authors seem to have made all of the code and raw files available, so it should be possible to re-train the networks from scratch, but that seems like an unnecessary deterrent to wider adoption/refining of the technique.

Typos

79: feature combination s (I’m not sure about this)

98: Sentence cuts off

905: vocalization..

Reviewer #2: USV – plos computational biology

The authors propose an interesting approach to sex discrimination in USV. They rightly identify that this a very challenging line of investigation.

Automated methods were used to automatically extracted USVs - please provide validation data on these methods as automated extraction is not a straightforward process and rarely as accurate as claimed.

In presenting the final outcomes, it would be helpful to report the metrics resulting from the rerun analysis as this more likely represents what would be observed in a novel cohort. Ie. performance is reduced.

Re-analyzing previously published data which reported no differences between groups, and then reporting different outcomes, is an important process.

Wording of this phrase needs revising “As intruders we used anaesthetized females to ensure that only the resident mouse could emit calls.”

More information on the WT/Cortexless paradigm is needed in the main document rather than making the reader find it in the supplementary materials.

The authors mention that data were derived from a subset of both cohorts. How were these subsets selected? Were only the ‘best ‘recordings selected, was there any bias in this process or were they randomly selected?

The number of calls appears sufficient but is 17 representative of mice in general?

DNN are a great way forward in this field, but as in other fields, understanding the machinations within the convolution and connected layers and how the output represents a concrete concept is challenging.

**Have all data underlying the figures and results presented in the manuscript been provided?**

Reviewer #1: Yes

Reviewer #2: Yes

PLOS authors have the option to publish the peer review history of their article (what does this mean?). If published, this will include your full peer review and any attached files.

Reviewer #1: Yes: Kevin R Coffey

Reviewer #2: No

---

## [Editor Report · Decision Letter 1]

12 Mar 2020

Dear Mr Ivanenko,

Thank you very much for submitting your manuscript "Classifying sex and strain from mouse ultrasonic vocalizations using deep learning" for consideration at PLOS Computational Biology. As with all papers reviewed by the journal, your manuscript was reviewed by members of the editorial board and by several independent reviewers. The reviewers appreciated the attention to an important topic. Based on the reviews, we are likely to accept this manuscript for publication, providing that you modify the manuscript according to the review recommendations.

Dear Alexander and Bernhard,

As you know from our previous correspondence you have performed an interesting analysis and showed that the USV in mice show some degree of sexual dimorphism but that this is exhibited in a complex acoustic signature that can best be extracted from a spectrographic representation. I still have some comments that I would like you to address. The mostly relate to your methods. You also use descriptions and language that are not completely in-line with what bio-acousticians use. I like your analyses of your networks.

Minor comments:

1. Methods: “Spectrograms were converted to a sparse representation by setting all values below 70th percentile to zero, which eliminated most close to silent background noise bins”. It is common to talk about thresholds in dB. So I suggest stating the corresponding dB threshold (e.g. This floor corresponds to a xx dB thresholds below peak).

2. “Hence, vocalizations longer than 100 ms (11.7%) were included shortened.” Reword: “Hence, vocalizations longer than 100 ms () were truncated to 100…”. (I assume the end was deleted – otherwise use the correct words to explain how it was shortened).

3. I know that you already provided more details on the NN but I am here also asking you to provide more details for the calculation of the auditory features. You should also use descriptions (and terms) that are well understood among bioacousticians. I am assuming that [0,1] means any value between 0 and 1? You figure 1 seems to say that these are attributed by visual inspection (by hand) but only 1, 0.5 and 0???:

a. E.g. how is the broadband defined? And how do you go from 0 to 1? Why not use Weiner Entropy which might be more common in sound analyses?

b. Similarly for complexity? Is this the Weiner entropy?

c. Similarly for tremolo? Specific the equation that you used.

d. What is the average Frequency of a vocalization? The spectral mean? Or the average fundamental? These are very different. You should probably have both. You do talk about the mean spectral energy for detecting vocalizations. If that is what you use, use the same term.

e. Spectral mean requires a distribution. How is this one estimated?

f. Also Power requires a frequency range – just all energy above 25 kHz. Normalized by duration? dB is a unitless measure (relative) so dB2 is meaningless. Both amplitude and power can be expressed in dB.

4. I don’t think that any of the measures should be assigned by hand. Ok for Direction, Peaks and Breaks but Broadband and Complexity should be quantified. Why not use measures such as spectral bandwidth and Weiner entropy? You could also do something like pitch saliency.

5. I still find your choice of acoustic features somewhat limited. I think that some of the most experienced bioacousticians might wince and I would not like that given that my name will also be on your report as the associate editor. I would also add fundamental. You could do the mean but also do a time-varying fundamental (that you can express with PC for dimensionality reduction if you want). My lab as a Python toolbox that you can find in github.com:/theunissenlab/BioSoundTutorial that you might find useful to efficiently extract features. But don’t take my word for it – you can also see what has been published by other groups. One issue is that mice USV are late in the game and that most folks that have analyzed them are not bioacousticians so you will need to get inspired by analyses in other animals.

Sincerely,

Frédéric E. Theunissen

Associate Editor

PLOS Computational Biology

Lyle Graham

Deputy Editor

PLOS Computational Biology

[LINK]

Dear Alexander and Bernhard,

As you know from our previous correspondence you have performed an interesting analysis and showed that the USV in mice show some degree of sexual dimorphism but that this is exhibited in a complex acoustic signature that can best be extracted from a spectrographic representation. I still have some comments that I would like you to address. The mostly relate to your methods. You also use descriptions and language that are not completely in-line with what bio-acousticians use. I like your analyses of your networks.

Minor comments:

1. Methods: “Spectrograms were converted to a sparse representation by setting all values below 70th percentile to zero, which eliminated most close to silent background noise bins”. It is common to talk about thresholds in dB. So I suggest stating the corresponding dB threshold (e.g. This floor corresponds to a xx dB thresholds below peak).

2. “Hence, vocalizations longer than 100 ms (11.7%) were included shortened.” Reword: “Hence, vocalizations longer than 100 ms () were truncated to 100…”. (I assume the end was deleted – otherwise use the correct words to explain how it was shortened).

3. I know that you already provided more details on the NN but I am here also asking you to provide more details for the calculation of the auditory features. You should also use descriptions (and terms) that are well understood among bioacousticians. I am assuming that [0,1] means any value between 0 and 1? You figure 1 seems to say that these are attributed by visual inspection (by hand) but only 1, 0.5 and 0???:

a. E.g. how is the broadband defined? And how do you go from 0 to 1? Why not use Weiner Entropy which might be more common in sound analyses?

b. Similarly for complexity? Is this the Weiner entropy?

c. Similarly for tremolo? Specific the equation that you used.

d. What is the average Frequency of a vocalization? The spectral mean? Or the average fundamental? These are very different. You should probably have both. You do talk about the mean spectral energy for detecting vocalizations. If that is what you use, use the same term.

e. Spectral mean requires a distribution. How is this one estimated?

f. Also Power requires a frequency range – just all energy above 25 kHz. Normalized by duration? dB is a unitless measure (relative) so dB2 is meaningless. Both amplitude and power can be expressed in dB.

4. I don’t think that any of the measures should be assigned by hand. Ok for Direction, Peaks and Breaks but Broadband and Complexity should be quantified. Why not use measures such as spectral bandwidth and Weiner entropy? You could also do something like pitch saliency.

5. I still find your choice of acoustic features somewhat limited. I think that some of the most experienced bioacousticians might wince and I would not like that given that my name will also be on your report as the associate editor. I would also add fundamental. You could do the mean but also do a time-varying fundamental (that you can express with PC for dimensionality reduction if you want). My lab as a Python toolbox that you can find in github.com:/theunissenlab/BioSoundTutorial that you might find useful to efficiently extract features. But don’t take my word for it – you can also see what has been published by other groups. One issue is that mice USV are late in the game and that most folks that have analyzed them are not bioacousticians so you will need to get inspired by analyses in other animals.
---

## [Editor Report · Decision Letter 2]

30 Apr 2020

Dear Mr Ivanenko,

We are pleased to inform you that your manuscript 'Classifying sex and strain from mouse ultrasonic vocalizations using deep learning' has been provisionally accepted for publication in PLOS Computational Biology.

Best regards,

Frédéric E. Theunissen

Associate Editor

PLOS Computational Biology

Lyle Graham

Deputy Editor

PLOS Computational Biology

dear Bernhard,

Thank you for your patience and carefully addressing all of my questions and recommendations.

Best wishes,

F

---

## [Editor Report · Acceptance letter]

4 Jun 2020

PCOMPBIOL-D-19-01759R2 

Classifying sex and strain from mouse ultrasonic vocalizations using deep learning

Dear Dr Englitz,

I am pleased to inform you that your manuscript has been formally accepted for publication in PLOS Computational Biology. Your manuscript is now with our production department and you will be notified of the publication date in due course.

With kind regards,

Laura Mallard
